# A Multi-Year, Multi-Cultivar Approach to Differential Expression Analysis of High- and Low-Protein Soybean (*Glycine max*)

**DOI:** 10.3390/ijms24010222

**Published:** 2022-12-23

**Authors:** Julia C. Hooker, Nour Nissan, Doris Luckert, Martin Charette, Gerardo Zapata, François Lefebvre, Ramona M. Mohr, Ketema A. Daba, Thomas D. Warkentin, Mehri Hadinezhad, Brent Barlow, Anfu Hou, Ashkan Golshani, Elroy R. Cober, Bahram Samanfar

**Affiliations:** 1Agriculture and Agri-Food Canada, 960 Carling Ave, Ottawa, ON K1A 0C6, Canada; 2Department of Biology, Ottawa Institute of Systems Biology, Carleton University, 1125 Colonel By Dr., Ottawa, ON K1S 5B6, Canada; 3Canadian Centre for Computational Genomics, 740 Dr. Penfield Ave, Montréal, QC H3A 0G1, Canada; 4Agriculture and Agri-Food Canada, 2701 Grand Valley Road, Brandon, MB R7A 5Y3, Canada; 5Crop Development Centre, University of Saskatchewan, Saskatoon, SK S7N 5A8, Canada; 6Agriculture and Agri-Food Canada, Morden, MB R6M 1Y5, Canada

**Keywords:** *Glycine max*, glycinin (11S), conglycinin (7S), seed protein content, differential gene expression, transcriptome-wide analysis

## Abstract

Soybean (*Glycine max* (L.) Merr.) is among the most valuable crops based on its nutritious seed protein and oil. Protein quality, evaluated as the ratio of glycinin (11S) to β-conglycinin (7S), can play a role in food and feed quality. To help uncover the underlying differences between high and low protein soybean varieties, we performed differential expression analysis on high and low total protein soybean varieties and high and low 11S soybean varieties grown in four locations across Eastern and Western Canada over three years (2018–2020). Simultaneously, ten individual differential expression datasets for high vs. low total protein soybeans and ten individual differential expression datasets for high vs. low 11S soybeans were assessed, for a total of 20 datasets. The top 15 most upregulated and the 15 most downregulated genes were extracted from each differential expression dataset and cross-examination was conducted to create shortlists of the most consistently differentially expressed genes. Shortlisted genes were assessed for gene ontology to gain a global appreciation of the commonly differentially expressed genes. Genes with roles in the lipid metabolic pathway and carbohydrate metabolic pathway were differentially expressed in high total protein and high 11S soybeans in comparison to their low total protein and low 11S counterparts. Expression differences were consistent between East and West locations with the exception of one, *Glyma.03G054100*. These data are important for uncovering the genes and biological pathways responsible for the difference in seed protein between high and low total protein or 11S cultivars.

## 1. Introduction

Soybean (*Glycine max* (L.) Merr.) is among the world’s most important crops due to the widespread uses for its seed protein and oil. By 2050 the global population is expected to approach 10 billion people, requiring the implementation of more efficient and imperishable food production and storage strategies [1]. Soybean has the capacity to fix inert atmospheric nitrogen into more useful biologically available forms within root nodules via symbiosis with rhizobia in the soil [2]. This unique relationship incentivizes the incorporation of soybean into crop rotation practices because it reduces the need to add nitrogen fertilizers, which are costly and have a role in greenhouse gas emissions of nitrous oxide [3]. Soybean plays an important role in improving soil quality and thereby serves as a pillar in strategic planning of sustainable agriculture.

Seed protein and oil content are complex quantitatively inherited traits that are influenced by genotype, environment and the interaction between the two [4,5]. A strong direct and/or indirect phenotypic correlation is evident in the inverse balance of these two storage components [6]. When considering soybean seed storage, an important balance exists between proteins, lipids and carbohydrates and it is important to consider expression of genes related to each of these macromolecules. Carbohydrate metabolism is the precursor to protein and oil biosynthesis, making it a key decision-making step in downstream seed storage molecule biosynthesis. Sugar transporters have been identified as playing a role in protein content in soybeans [7,8]. Seed protein content accounts for 30–46% (average 40%) and oil accounts for 12–24% (average 20%) of the seed weight [6]. There is a limit to the current understanding of oil and storage protein regulation and accumulation in soybean seeds, as well as the determining factors for the allocation of carbon to the production of proteins or oil [9,10].

Soybean seeds, like many plant seeds, accumulate considerable amounts of storage proteins as a reserve of carbon, nitrogen and sulfur for the germinating/developing of seedlings [5,11]. Seed storage proteins are categorized by their sedimentation coefficient at 0.5 M ionic strength: 2S, 7S, 11S and 15S fractions [12]. The most abundant are the 11S (glycinins) fraction and the 7S (β-conglycinins) fraction, which account for approximately 40% and 30%, respectively [13]. Together the 11S and 7S globulins provide a total of 18 amino acids, which include the 10 essential amino acids, but the amount of sulfur-containing amino acids, cysteine (Cys) and methionine (Met), are not sufficient for the nutrient requirements of monogastric animals [14]. Excess amino acids are not stored by the body like fats and carbohydrates, so they must be obtained from daily food intake.

Glycinins normally form a 360 kDa hexamer made up of both acidic and basic subunits; β-conglycinins typically exist as a 180 kDa trimer of α, α′ and β subunits [4,15]. Glycinins have approximately 3–4 times the amount of sulfur-containing amino acids (Met, Cys) in comparison to β-conglycinins [4]. Evidently, the β subunit of β-conglycinin has poor nutritional value because of the lack of sulfur-containing amino acids. Through the promotion of accumulation of glycinins and simultaneous suppression of β-conglycinin accumulation, it is entirely possible to make valuable improvements to the nutritional quality of soybean by increasing the sulfur-containing amino acid content [4]. The seed protein composition, particularly that of the 11S and 7S globulins, is important for the germinating seed as well as the nutritional and functional quality of soy foods [4]. Subunit composition of both the glycinins and β-conglycinins, as well as the ratio of the two (i.e., 11S:7S ratio), affect the quality of the resultant food product [4]. Studies have shown that the 11S and 7S proteins present in soybean seed affect food sensory attributes including texture, taste, heat stability, tofu gelling and water retention [16]. With this information, the development of high-quality protein composition in seeds must be acknowledged when considering all aspects of improving soybean seed quality.

Currently, there are 15 known genes which encode β-conglycinins (*CG1*–*CG15*) and five known genes encoding glycinins (*Gy1*–*Gy5*) [17,18]. Gene mining for sulfur-containing amino acid metabolites in soybean predicted 12 candidate genes in addition to quantitative trait loci (QTLs) for Cys and Met content in 11S and 7S fractions on chromosomes 1, 3, 4, 6, 10, 13, 16, 17, 19 and 20 [19]. In another study, F6-derived recombinant inbred lines (RILs) were used in linkage mapping to identify three QTLs associated with glycinin on chromosomes 17, 19 and 20 and two QTLs associated with β-conglycinin on chromosomes 17 and 16 [15]. One QTL was associated with the α′ subunit of β-conglycinin on chromosome 10 and five QTLs associated with 11S subunits on chromosomes 3, 10, 13 and 19 [20].

Soybean production in Canada has observed a consistently lower (by ~1–5%) seed protein content in Western growing regions in comparison to Eastern growing regions [21]. In 2021, the average Eastern soybean protein content on a dry basis was 40.3%, while the Western-grown soybeans had an average seed protein content of 36.0% [21]. Compounded with the other challenges of growing soybean in western Canada (i.e., less precipitation, cooler temperatures and shorter growing season), the issue with lower-protein soybean contributes to a reduced interest in soybean production and subsequently the loss of a key nitrogen fixing option from Canadian crop rotation practices. Seed protein and oil composition are important factors in soybean seed quality; a deeper foundational understanding of the genes differentially expressed (DE) between high and low protein soybeans across Canada is valuable for tailored, high-quality soybean breeding.

The current understanding of seed storage protein regulation and accumulation in soybean is limited. Here we used RNA-seq and differential expression (DE) analysis of high and low total protein (TP) soybeans and high and low 11S soybeans, grown in multiple regions of Canada over three years, in order to uncover commonly differentially expressed genes which may influence seed protein and oil accumulation.

## 2. Results

### 2.1. Seed Content, Composition and Selection of Lines for DE

Soybean seed composition data from each line in each location was collected each year from 2018–2020. The average seed protein content for each line (1–10) was taken across all locations over 3 years in order to narrow down lines for DE analysis. Figure 1A shows the average seed TP and seed oil content for each line over 2018–2020. Appendix A includes the 3-year averages for TP, oil, 11S, 7S and 11S:7S for all 10 lines and the standard deviations from the means. Lines 8, 9 and 10 had the lowest average oil content and lines 2, 1, 6 and 3 had the first, second, third and fifth highest oil content, respectively (Figure 1A, Appendix A). Lines 1, 2 and 3 had the lowest protein content and lines 8, 9 and 10 had the highest protein content (Figure 1A, Appendix A). Based on these results lines 8, 9 and 10 were designated “high TP” and lines 1, 2 and 3 were designated “low TP” and were selected for DE analysis. Table 1 shows the seed protein and seed oil content for high TP (i.e., low oil) soybeans and low TP (i.e., high oil) soybean samples used for DE analysis (difference in TP content between high and low TP soybeans *p*-value 0.01). The low TP samples served as the control and the high TP are the experimental data; therefore, DE shows up/downregulation in the high TP soybeans.

Seed protein composition analysis was then considered for average 11S and average 7S content, as a percentage of total protein, for all lines. Figure 1B shows the proportion of 11S and 7S of the total seed protein content (primary y axis) and the 11S:7S ratio value (secondary y axis) for each line across all 3 years (2018–2020). Lines 1, 5 and 9 have the highest 7S content and lines 2, 4, 3 and 8 have the 4 lowest 7S content values, respectively (Figure 1B, Appendix A). The three lines with highest 11S content are lines 2, 4, 8 and three lines with lowest 11S content are lines 1, 5, 9 (Figure 1B, Appendix A). Incidentally, the lines with the lowest 11S:7S ratio were the same lines that had the lowest 11S content (lines 1, 5, 9). Lines 2, 4, 3 and 8 had the top four highest 11S:7S ratios, respectively (Figure 1B, Appendix A). Based on these results, lines 1, 5 and 9 were grouped as “low 11S” and lines 2, 4 and 8 were designated the “high 11S” group selected for DE analysis. The average 11S content, 7S content and ratio of 11S:7S for the high and low 11S soybean lines are listed in Table 1 (difference in 11S content between high and low 11S soybeans *p*-value 0.005). Low 11S soybeans were used as the control and high 11S soybeans were used as the experimental sample; therefore, when discussing DE, the log2FC up- and downregulation describe the DE in the high 11S samples.

### 2.2. RNA-seq and DE Output Quality

In total, 234 samples (10 year-locations, 8 lines) were used to create 20 DE datasets: 10 high TP vs. low TP and 10 high 11S vs. low 11S. Across all samples, Q30 scores of at least 36 were accepted. In total, across three replicates from lines 1, 2, 3, 4, 5, 8, 9, 10 in each location from 2018–2020 there were 5,267,207,213 reads. The TP analysis included a total of 3,932,862,305 reads across all years and locations combined. Analysis of high TP data came from 1,968,419,408 reads and low TP data came from 1,964,442,897 reads. The 11S analysis included 3,478,643,109 reads across all years and locations. The high 11S samples (lines 2, 4, 8) had a total of 2,030,526,766 reads across all years and locations; the low 11S (lines 1, 5, 9) had a total of 2,002,088,417 reads. The average paired read survival rate was 98.0%, with the lowest survival rate of 90.9%. Figure 2 shows the principal component (PC) analysis of the normalized RNA-seq read data for each high TP vs. low TP DE analysis. Figure 3 shows the PC analysis of the normalized RNA-seq read data for each high 11S vs. low 11S DE analysis. Because both Figure 2 and Figure 3 are built on normalized expression data, phenotypic TP and 11S data are not necessarily always the first PC. For both Figure 2 and Figure 3, orange data points represent high TP or 11S and blue data points represent low TP or 11S, respectively. The line numbers corresponding to each individual datapoint are presented next to each point.

### 2.3. High vs. Low Seed Protein Content DE

From each of the high vs. low (Lines 8, 9, 10 vs. Lines 1, 2, 3) TP DE analyses, the top 15 most upregulated and the 15 most downregulated genes and their respective log2foldchanges (log2FCs) in expression were extracted for each environment (300 genes total). Genes were sorted based on frequency of occurrence; those that were present in at least 5/10 DE location-year datasets were short-listed. Table 2 shows the shortlist of genes for the cumulative high vs. low TP analyses across all 10 DE datasets. Included in this table is the log2FC difference in expression in high TP soybeans for each candidate gene in each dataset and the total number of datasets out of 10 from which each gene was found to be in the top 15. Eleven shortlisted genes were upregulated in at least 5 of 10 high vs. low TP DE datasets; seven shortlisted genes were downregulated in at least 5 of 10 high vs. low TP DE datasets. Figure 4 shows the individual relative expression heatmaps for the shortlisted genes as they appear in each location-year DE dataset.

Three genes, *Glyma.03G057800*, *Glyma.10G092400* and *Glyma.16G081500*, were each identified to be upregulated in 8/10 high vs. low TP DE datasets (Table 2). The log2FC upregulation of *Glyma.03G057800* across the datasets ranged from 5.14 (Morden 2020) to 27.5 (Ottawa 2019), with an average log2FC of 16.8. In Figure 4, gene 1 in the heatmaps corresponds to *Glyma.03G057800*. Morden 2018 and Brandon 2019 are the two datasets which did not identify *Glyma.03G057800* within the most upregulated (nor downregulated) genes; in Morden 2018 *Glyma.03G057800* was the 93rd most upregulated gene (log2FC 0.99; Figure 4D) and in Brandon 2019 *Glyma.03G057800* was the 17th most upregulated gene (log2FC 6.75; Figure 4H). The log2FC upregulation of *Glyma.10G092400* ranged from 5.08 (Morden 2018) to 25.5 (Ottawa 2020), averaging 13.9 (Table 2). In Figure 4, gene 2 in the heatmaps corresponds to *Glyma.10G092400*. *Glyma.10G092400* is within the top 15 upregulated genes of all datasets with the exception of Ottawa 2018 and Saskatoon 2020; in Ottawa 2018, it was the 32nd most upregulated gene (log2FC 3.46; Figure 4A) in high TP soybeans, while in Saskatoon 2020 it was not identified to be DE (Figure 4J, non-DE shortlisted genes are shown in grey). *Glyma.16G081500* was among the topmost prevalent upregulated genes in the high vs. low TP DE analyses. In Figure 4, gene 3 in the heatmaps corresponds to *Glyma.16G081500*. This gene had a range of log2FC DE from 23.80 (Ottawa 2020; Figure 4C) to 31.83 (Morden 2020; Figure 4F) and an average log2FC of 29.40 in high TP soybeans. *Glyma.16G081500* was identified in our study to be among the top 15 genes in 8/10 datasets, only excluding Morden 2018 and Brandon 2018, in neither of which *Glyma.16G081500* was identified to be DE (Figure 4D,G).

An additional four genes from the high vs. low TP DE analyses were found to be upregulated in 7 of the 10 DE datasets; *Glyma.01G179100*, *Glyma.02G060600*, *Glyma.10G092300* and *Glyma.19G140200* (Table 2). In Figure 4, gene 4 in the heatmaps corresponds to *Glyma.01G179100*. The log2FC upregulation of *Glyma.01G179100* had a wide range from 2.54 (Saskatoon 2020; Figure 4J) to 24.7 (Brandon 2019; Figure 4H), with an average log2FC 11.6 (Table 2). In Morden 2018 *Glyma.01G179100* was the 933rd most upregulated gene, with a log2FC increase of 0.35 (Figure 4D). In Morden 2019 *Glyma.01G179100* was the 19th most upregulated gene, with a log2FC increase by 4.16 (Figure 4E). In Ottawa 2019 *Glyma.01G179100* was the 18th most upregulated gene, with a log2FC increase of 7.35 (Figure 4B). In Figure 4, gene 5 in the heatmaps corresponds to *Glyma.02G060600*. The log2FC DE of *Glyma.02G060600* in high TP soybeans ranges from 26.7 (Morden 2018; Figure 4D) to 35.9 (Ottawa 2019; Figure 4B) and the average DE was by log2FC 32.9 (Table 2). *Glyma.02G060600* was not DE between low and high TP soybeans in Morden 2020, Brandon 2018 and Saskatoon 2019 (Figure 4G,F,I). In Figure 4, gene 6 in the heatmaps corresponds to *Glyma.10G092300*. *Glyma.10G092300* was within the top 15 upregulated genes in 7/10 high TP DE analyses, with a minimum log2FC increase by 4.84 (Morden 2018; Figure 4D), a maximum log2FC by 27.1 (Ottawa 2020; Figure 4C) and an average log2FC of 20.3 (Table 2). *Glyma.10G092300* was the 22nd most upregulated gene in high TP soybeans in Brandon 2018 (Figure 4G); it was the 17th most upregulated gene in high TP soybeans in Morden 2019 (Figure 4E); and was not detected to be DE between high and low TP soybeans in Saskatoon 2020 (Figure 4J). In Figure 4, gene 7 in the heatmaps corresponds to *Glyma.19G140200*. *Glyma.19G140200* was upregulated in 7 of the 10 datasets by a minimum log2FC 18.4 (Brandon 2018; Figure 4G), a maximum log2FC 22.8 (Ottawa 2019; Figure 4B) (Table 2), with an average increase by log2FC 21.2. *Glyma.19G140200* was not DE in high TP soybeans in Ottawa 2018, Ottawa 2020 and Morden 2018 (Figure 4A,C,D).

*Glyma.13G077600* was identified to be within the top 15 downregulated genes in 7/10 high vs. low TP DE datasets (Table 2). In Figure 4, gene 12 in the heatmaps corresponds to *Glyma.13G077600*. The log2FC DE of *Glyma.13G077600* ranged from −13.4 (Brandon 2018) to –35.4 (Ottawa 2019) and an average log2FC of −31.2 across the seven datasets (Table 2). *Glyma.13G077600* was not identified to be DE in high TP soybeans grown in Morden 2019, Brandon 2019, nor Saskatoon 2020 (Figure 4E,H,J). Gene 13 in the heatmaps in Figure 4 corresponds to *Glyma.15G246500*. *Glyma.15G246500* was downregulated in 6/10 high vs. low TP DE analyses. The minimum log2FC difference in expression of *Glyma.15G246500* ranged from −19.2 (Morden 2018) to −26.3 (Ottawa 2019), averaging at −23.2 (Table 2). *Glyma.15G246500* was not identified to be DE in high TP soybeans in Brandon 2018, Brandon 2019, Saskatoon 2019 and Saskatoon 2020 (Figure 4G–J). Gene 16 in the heatmaps in Figure 4 corresponds to *Glyma.03G054100*. *Glyma.03G054100* is downregulated in 5/10 high vs. low TP datasets; downregulation ranges from log2FC –9.52 (Brandon 2019; Figure 4H) to –25.9 (Saskatoon 2019; Figure 4I) and an average of −21.6 (Table 2). *Glyma.03G054100* was the 67th most downregulated gene in Morden 2018, by a log2FC −1.08 (Figure 4D); it was not DE in high TP soybeans in Morden 2019, Morden 2020, Brandon 2018 and Saskatoon 2020 (Figure 4E–G,J).

### 2.4. High vs. Low 11S Content DE

From each of the 10 year-location high vs. low 11S (lines 2, 4, 8, vs. 1, 5, 9) DE analyses, the top 15 most upregulated and the 15 most downregulated genes and their respective log2FCs were extracted (300 genes total). Genes were sorted based on frequency of occurrence and those that were present in at least 5/10 DE datasets were short-listed. Three genes, *Glyma.01G016700*, *Glyma.06G306900*, *Glyma.19G231100* and *Glyma.16G086800*, were DE in 5 or 6 of 10 datasets; however, in some instances these genes were upregulated and in some instances they were downregulated. This inconsistency rules them out as a genes of interest in this study. Table 3 shows the shortlist of the most frequently occurring genes across the 10 11S DE datasets. There are eight genes upregulated in at least 5/10 DE datasets; *Glyma.19G084500*, *Glyma.02G077300*, *Glyma.17G209900*, *Glyma.01G091300*, *Glyma.06G287800*, *Glyma.10G141200*, *Glyma.14G204900* and *Glyma.18G112500* (Table 3). There are six genes downregulated in at least 5/10 datasets; *Glyma.13G077600*, *Glyma.17G261800*, *Glyma.01G127800*, *Glyma.03G054100*, *Glyma.12G156500* and *Glyma.18G082700*. Figure 5 shows the relative expression heatmaps of each of the shortlisted genes as they appear in each year-location 11S DE analysis.

*Glyma.19G084500* was within the top 15 most upregulated genes across 9/10 DE datasets. Gene 1 in the heatmaps in Figure 5 corresponds to *Glyma.19G084500*. The log2FC DE in high 11S soybeans of *Glyma.19G084500* ranged from 21.5 (Saskatoon 2019; Figure 5I) to 27.5 (Morden 2020; Figure 5F), with an average upregulation by a log2FC of 24.8 (Table 3). Brandon 2018 is the only dataset in which *Glyma.19G084500* did not fall within the top 15 upregulated (nor downregulated) genes; further, it was not identified to be DE at a *p*-value < 0.05 in Brandon 2018 (Figure 5G).

*Glyma.02G077300* was identified within the top 15 most upregulated genes in 8/10 high vs. low 11S DE analyses (Table 3). In Figure 5, gene 2 corresponds with *Glyma.02G077300*. Upregulation in high 11S soybeans ranged from 16.1 (Morden 2019; Figure 5E) to 22.2 (Ottawa 2020; Figure 5C), with an average log2FC of 20.3. Morden 2018 and Ottawa 2018 were the only high vs. low 11S datasets that did not identify *Glyma.02G077300* within the 15 most up- or downregulated genes (Table 3). In Morden 2018 *Glyma.02G077300* was upregulated by a log2FC of 3.28, the 32nd most upregulated gene in the dataset (Figure 5D). *Glyma.02G077300* was not identified to be DE (*p*-value < 0.05) between high and low 11S soybeans in Ottawa in 2018 (Figure 5A).

*Glyma.13G077600* was found to be among the most persistently downregulated genes in high 11S soybeans compared to low 11S soybeans (Table 3). In Figure 5, gene 9 corresponds with *Glyma.13G077600*. *Glyma.13G077600* was downregulated in 7/10 of the DE datasets; log2FC DE ranged from −29.6 (Saskatoon 2020; Figure 5J) to −33.5 (Ottawa 2019; Figure 5B) and the average downregulation of *Glyma.13G077600* was by a log2FC of −32.0. In Brandon 2018, *Glyma.13G077600* was downregulated by a log2FC of −9.95, but not within the top 15 most downregulated genes (Figure 5G). *Glyma.13G077600* was not identified to be DE between high and low 11S soybeans from Ottawa 2020 and Morden 2019 (Figure 5C,E).

*Glyma.17G261800* was also downregulated in 7/10 DE datasets between high vs. low TP soybeans (Table 3); log2FC DE ranged from −15.0 (Brandon 2018; Figure 5G) to −46.0 (Morden 2019; (Figure 5E) and the average was −38.4. In Figure 5, gene 10 corresponds with *Glyma.17G261800*. *Glyma.17G261800* was not identified within the top 15 most downregulated (nor upregulated) genes in Ottawa 2019, Morden 2020 and Brandon 2019 (Figure 5B,F,H). *Glyma.17G261800* is the 17th most downregulated gene (log2FC −4.33) in Brandon 2019 high TP soybeans (Figure 5H). In Ottawa 2019 *Glyma.17G261800* was the 18th most downregulated gene (log2FC −5.56; Figure 5B). In Morden 2020 *Glyma.17G261800* was downregulated by a log2FC of −1.58, the 67th most downregulated gene in the Morden 2020 dataset (Figure 5F).

The annotated list of up- and downregulated shortlist genes from both differentially expressed in high TP soybeans and high 11S soybeans are listed in order of chromosome position in Appendix A. Appendix A also includes the potential role of each gene in relation to seed content accumulation, the orientation of DE and the data from which each gene was found to be significant.

### 2.5. Gene Ontology

Appendix A lists all the biological process (BP) and molecular function (MF) gene ontologies (GOs) for all shortlisted genes. In order to gain a broad view of the relationships of the BP GOs associated with all shortlisted genes, the upregulated shortlist and downregulated shortlist were run through the SoyBase GO Term Enrichment Tool [22] (soybase.org (accessed on 17 May 2022)). The GO terms were submitted to Revigo [23,24] (http://revigo.irb.hr/ (accessed on 17 May 2022)), using *A. thaliana* as a reference species to assess relative similarity between GO terms. Figure 6 shows the relationship between indispensable (dispensability score of 0 on a scale of 0 to 1) BP terms associated with the high TP upregulated (Figure 6A) and downregulated (Figure 6B) short list of genes. Indispensable terms from the upregulated genes in high TP soybeans are oligopeptide transport (GO:0006857), cell population proliferation (GO:0008283), response to xenobiotic stimulus (GO:0009410), regulation of G2/M transition of mitotic cell cycle (GO:0010389) and floral organ formation (GO:0048449). Indispensable terms from the downregulated genes in high TP soybeans are regulation of transcription–DNA-templated (GO:0006355), regulation of gene expression (GO:0010468), phosphatidylinositol biosynthetic process (GO:0006661), embryo development ending in seed dormancy (GO:0009793) and defense response to bacterium (GO:0042742). Figure 7 shows the relationship between BP GO terms associated with the high 11S upregulated (Figure 7A) and downregulated (Figure 7B) shortlisted genes. Appendix A provides the Revigo outputs including each GO term and the corresponding frequency, uniqueness and dispensability and PC analysis. Importantly, the high TP soybeans upregulating lipid metabolic process (GO:0006629) and carbohydrate metabolic process (GO:0005975) had the two highest logSize values on the Revigo plot (Figure 7A) of 3.04 and 3.05, respectively and the highest frequency values of 5.08 and 5.19, respectively (Appendix A). The dispensability score for lipid metabolic process was 0.102 and the dispensability score for carbohydrate metabolic process is 0.189, indicating removal of any of the multiple terms that are daughter terms under the lipid metabolic process (GO:0006629) and carbohydrate metabolic process (GO:0005975) umbrellas would largely impact the overall relationship structure (and PCs). The indispensable BP GO terms from the genes upregulated in high 11S soybeans are response to wounding (GO:0009611), photorespiration (GO:0009853) and leaf senescence (GO:0010150) (Figure 7A, Appendix A). Other important terms upregulated in high 11S soybeans include glucose metabolic process (GO:0006006), starch metabolic process (GO:0005982), maltose catabolic process (GO:0000023) and oxylipin metabolic process (GO:0031407) (Figure 7A, Appendix A). The indispensable BP GO terms from the downregulated genes in high 11S soybeans are gibberellic acid mediated signaling pathway (GO:0009740) and seed dormancy process (GO:0010162) (Figure 7B, Appendix A).

### 2.6. East vs. West Analysis

Division of the shortlist data on the basis of location shows some discrepancies in expression of some key genes between soybeans grown in East (Ottawa) versus West (Morden, Brandon, Saskatoon) Canada. The average DE for each shortlisted gene was taken from Ottawa (2018–2020) and the three West locations combined (2018–2020) for both high vs. low TP and high vs. low 11S soybeans samples. Appendix A shows the average DE for each short-listed gene in the East and West for both high vs. low TP soybeans and high vs. low 11S soybeans.

Between the high vs. low TP soybeans grown in East and West, the eastern samples showed much higher expression of *Glyma.03G057800*; eastern-grown soybeans had an average DE of log2FC 25.2 compared to log2FC 11.9 in the West (standard deviation of 9.31). *Glyma.10G092400* was also found to be much more upregulated in eastern-grown soybeans than the western-grown soybeans; the average upregulation in the East was log2FC 24.2, whereas the average upregulation in the West was log2FC 10.5 (standard deviation of 9.7).

Among the high vs. low 11S soybean samples, the average DE of each shortlisted gene is fairly similar between East and West, with the exception of *Glyma.03G054100*. *Glyma.03G054100* is downregulated in high 11S soybeans in the East by an average of log2FC −7.29, in the West this gene is downregulated by log2FC −38.9 (standard deviation of 22.4). This gene is also more downregulated in high TP soybeans in the West (log2FC –18.6 in the East; log2FC −25.2 in the West; standard deviation of 4.71).

## 3. Discussion

*Glyma.16G081500* is among the topmost upregulated genes in the high vs. low TP DE analyses (Table 2). *Glyma.16G081500* is uncharacterized in *G. max*, but the top BLASTP hit identified a Subtilisin-like serine protease in *Medicago truncatula* (Appendix A). Subtilase (SBT) family proteins have been identified in plants to play a broad range of biological functions involved in many different aspects of plant life, starting with seed and fruit development, cell wall modification, response to abiotic and biotic stressors, protein turnover, peptide growth factors, epidermal development and programmed cell death [25]. These functions are all favorable during seed development; allowing seed cell growth via cell wall modifications and cell-to-cell communication for choreographed development. Upregulation of *Glyma.16G081500* suggests that high TP soybeans may be better adapted for seed development than low TP soybeans, in part due to increased SBT function during seed development.

*Glyma.16G060600* is in the shortlist of upregulated genes in high TP soybeans (Table 2). *Glyma.16G060600* is uncharacterized in *G. max* but has the MF GO phospholipase activator activity (GO: 0016004) (Appendix A). The top BLASTP hit identified the ADP-ribosylation factor (Arf) in *M. truncatula* as the most closely related protein (Appendix A). The Arf family is composed of small GTP-binding proteins that play roles in intracellular trafficking and cargo sorting in yeast, animal cells and plant cells [26]. During seed development, precursors for seed storage proteins (in particular 11S and 7S proteins) are first synthesized in the ER. Arf proteins subsequently transport these protein subunits to protein precursor-accumulating vesicles (PSVs) by the Golgi-dependent trafficking pathway in which they are converted to mature subunits and accumulated in the developing seed. Arf1 is thought to also be involved in cargo protein sorting [26]. Arf1 is specifically shown to function in retrograde trafficking from the Golgi apparatus to the ER, as well as from the trans-Golgi network to the endosome [27]. In pea plants, storage proteins such as vicilin (7S) were identified to be sorted to the storage vacuole at the cis-Golgi [28]. In pumpkin seeds, GTP-binding proteins have been shown to target and/or fuse with the precursor-accumulating vesicles which accumulate and transport insoluble components of storage proteins to vacuoles directly from the ER [29]. The upregulation of *Glyma.16G060600* in high TP soybeans suggests increased Arf function in comparison to low TP soybeans, likely indicating increased transportation and sorting of storage protein precursors to PSVs for biosynthesis of storage proteins. This increased expression of Arf-like trafficking proteins for protein storage vacuoles in high TP soybeans is a plausible means for an underlying factor for the difference in TP between experimental soybean samples.

*Glyma.16G082200* is upregulated across high TP soybeans (Table 2). The top TAIR10 identity for *Glyma.16G082200* is the NAD(P)-binding Rossmann-fold superfamily proteins, also known as enoyl-acyl carrier protein (ACP)-dependent reductases (NCBI gene ID: 815152) (Appendix A). Enoyl-ACP reductase proteins are an enzymatic component of the mitochondrial type II fatty acid synthase pathway and plastidic type II fatty acid synthase activities, including acyl precursor synthesis for lipoic acid biosynthesis [30]. Upregulation of genes for Enoyl-ACP reductases suggests that these high TP soybeans are increasing biosynthesis of lipoic acid, essential for oxidation of carbohydrates among other important cell functions. *Arabidopsis* enoyl-ACP reductase knockdown mutants showed symptoms consistent with plants with deficient mitochondrial fatty acid synthase activities: depleted lipoliation of photorespiratory glycine cleavage system H-protein, glycine hyperaccumulation and reduced growth with very small ariel organs [30]. *Glyma.16G082200* was upregulated across high TP soybeans compared to low TP soybeans (Table 2). Reduced activity of enoyl-ACP reductases in low TP soybeans may reflect the symptoms observed in the *Arabidopsis* mutants reported by [30]. The phenotype of reduced aerial organs results in less photosynthetic tissue and therefore a reduced capacity for energy production and successful maturation. This gene may pinpoint a specific difference in carbohydrate metabolic pathways between high TP and low TP soybeans.

*Glyma.10G092300* was upregulated among high TP soybeans (Table 2). The top BLASTP hit for *Glyma.10G092300* is the Peptide transporter (PTR) 3-A from *M. truncatula* (Appendix A). PTRs are part of a larger umbrella family of nitrate transporter 1/peptide transporters (NTR/PTR; now known as the NPF family); a family characterized by their role in nitrogen uptake and transportation of nitrates and peptides [31]. Studies in rice have confirmed positive relationships between NPF expression and enhanced nitrogen allocation, nitrogen use efficiency, grain yield, branching, influx/concentration of nitrate and ammonium in the roots and potential kinase involvement [32]. Increased expression of *Glyma.10G092300* in high TP soybeans suggests enhanced allocation and use of nitrogen in these plants and subsequently likely attributes of the difference in TP.

Just downstream from *Glyma.10G092300*, *Glyma.10G092400* was also found to be upregulated across high TP soybeans (Table 2). With no lysine (WNK) kinase 3 is the top *Arabidopsis* homolog for *Glyma.10G092400*, which is otherwise uncharacterized in soybean (Appendix A). GmWNK1, a soybean WNK, is suggested to play a role in ABA signaling and ABA-mediated homeostatic response to osmotic changes in roots which mediates root architecture [33]. Both *Glyma.10G092300* and *Glyma.10G092400* are upregulated in all the same year-locations: Brandon-2018, Morden-2018, Brandon-2019, Ottawa-2019, Saskatoon-2019, Morden-2020 and Ottawa-2020 (Table 2). The only difference was found in Morden-2019 where *Glyma.10G092400* which was upregulated but not *Glyma.10G092300* (Table 2). These similar expression patterns may indicate a potential relationship between the roles of *Glyma.10G092300* (an NPF-like protein, with potential for kinase involvement) and *Glyma.10G092400* (a kinase). A relationship between the functional activities of *Glyma.10G092300* and *Glyma.10G092400* could be indicative of the crosstalk between ABA signaling and nitrogen availability.

*Glyma.09G184300*, upregulated in high TP soybeans, is identified as a bZIP transcription factor in *G. max* and TAIR10 identified a TGACG motif-binding factor 6 as the top *Arabidopsis* homolog (Table 2 and Appendix A). bZip transcription factors influence plant development, drought stress response, defense response and seed development. In *Phaseolus* bZip factors Regulator of MAT1 (ROM1) moderates lectin and storage protein gene transcription and expression is developmentally regulated (abundant in early embryogenesis and decreasing as maturation progresses) [34]. The increased expression of *Glyma.09G184300* in high TP soybeans, particularly during the R5 stage at which sampling took place, may make it a key gene that needs temporal regulation for improved seed storage protein accumulation. Temporal DE analysis of this transcription factor might be of interest to gain further insight to its role in seed development and protein content, including the gene(s) it acts upon.

*Glyma.03G057800*, a gene found to be highly upregulated in high TP soybeans, is predicted to be a rhodanese-like domain-containing protein in *G. max*. The top BLASTP hit identified as a Rhodanese-like family protein-like protein in *M. truncatula* and the TAIR10 hit identifies a Rhodanese/Cell cycle control phosphatase superfamily protein in *A. thaliana* (Appendix A). Proteins with Rhodanese (thiosulfate:cyanide sulfur-transferase) domains are versatile sulfur carriers capable of fulfilling reactive roles in metabolic and regulatory pathways, including senescence [35]. This may suggest that cell cycle control is differently regulated in high TP soybeans compared to low TP soybeans. High TP proteins may be expressing these genes for tightly controlled growth and development and/or low TP soybeans may be deficient in cell cycle control, leading to poorer growth and development under the same environmental conditions.

*Glyma.06G205700* was found to be downregulated across high TP soybeans (Table 2). The top BLASTP hit identified this gene as closely related to the Squamosa promoter-binding protein-like (SPL) transcription factor family protein (fragment) in ancestral soybean, *Glycine soja* (Appendix A). Squamosa family transcription factors have important roles regulating plant growth, development, response to stress, architecture and yield [36]. There is an inverse relationship between high protein and yield in soybean [37]; downregulation of *Glyma.06G205700* in high TP soybean may be important for improving protein accumulation in the seed and a compensatory response to increased protein content in seeds.

*Glyma.15G246500* was among the most commonly downregulated genes in high TP soybeans, present in the topmost downregulated genes in 6/10 high vs. low TP DE datasets. *Glyma.15G246500* is predicted as an uncharacterized protein LOC100812621 isoform X3 in *G. max*; BLASTP and TAIR10 results indicate the protein embryo defective 3012 (EMB3012) in *A. thaliana* to be the most closely related known protein (Appendix A). EMB family proteins have a diverse range of identified functions, but the majority are pentatricopeptide repeat (PPR) proteins, important for regulation of gene expression at the RNA level by facilitating post translational modifications such as splicing, editing and RNA stability [38]. EMB-defective *Arabidopsis* mutants produced seeds with defective embryo patterning, enlarged endosperm nuclear size, arrested or weak embryos [39]. The downregulation of *Glyma.15G246500* in high TP soybeans implies there is a difference in chromosome maintenance and/or regulation of gene expression that may play a role underlying the difference in seed protein content between high TP and low TP soybean varieties. Differential mRNA processing (i.e., splicing) between high TP and low TP soybeans is likely. Low TP soybeans may be compensating for their less adequate storage protein genetics (in comparison to high TP counterparts) by increasing their expression of *Glyma.15G246500*. Splice variance expression comparisons (differential transcript usage) between *Glyma.15G246500* in high TP soybeans and low TP soybeans would make an interesting next step to determine the effects of post translational modifications.

*Glyma.19G084500* was the most commonly occurring upregulated gene in high 11S soybeans, present in the top 15 upregulated genes from 9/10 high vs. low 11S DE analyses (Table 3). *Glyma.19G084500* is predicted to be a 52 kDa repressor of the inhibitor of the protein kinase-like in *G. max* with no known GOs (Appendix A). The top BLASTP hit identifies a hAT family dimerization domain containing protein in *Medicago truncatula* as the most closely related protein and the TAIR10 *Arabidopsis* homolog is a general transcription factor 2-related zinc finger protein (Appendix A). PFAM results identified this gene to have a hAT family C-terminal dimerization region (Appendix A). The *Activator* superfamily (hAT element superfamily) is a family of small and autonomous transposases with characteristic, highly conserved regions at the C-termini, important for dimerization. It would be of interest to investigate PPI prediction analysis and pulldown assays to uncover interacting partners for the product of *Glyma.19G084500*; this would be useful in further investigating the role, if any, this gene product plays in seed content accumulation.

*Glyma.17G209900* is among the topmost commonly upregulated genes in high 11S soybeans (Table 3). *Glyma.17G209900* is uncharacterized in *G. max* but has a BP GO for lipid metabolic process (GO:0006629) (Appendix A). The top BLASTP hit identified 12-oxo-phytodienoic acid reductase (OPR) in *Zea mays* as the most closely related protein and the TAIR10 hit identified 12-oxophytodienoate reductase 1 (OPR1) in *Arabidopsis* as a homolog (Appendix A). 12-oxophytodienoate reductase 1 (OPR1) along with OPR2 and OPR3 are isoenzymes of the 12-oxophytodienoate reductase [40]. These three enzymes are highly related genes of oxylipin 12-oxophytodienoic acid (OPDA) responsible for the synthesis of jasmonic acid (JA) [41]. JA and its derivatives regulate gene expression, influencing a spectrum of developmental processes including seed germination, root development, fertility, fruit ripening, plant defense and senescence. OPR1, OPR2 and OPR3 can all play a role in the synthesis of JA; however OPR3 was identified as the most efficient stereoisomer [40]. OPR3 catalyzes the reduction of 9S,13S-12-oxo-phytodienoate, which is responsible for the plant hormone JA. Mutations in the OPR3 lead to a loss of enzymatic activity, which in turn leads to trouble synthesizing JA and fine-tuning gene expression in plants. Our data indicates high 11S soybeans increased expression of *Glyma.17G209900* compared to low 11S soybeans. The effects of JA on protein and amino acid accumulation in oilseed crops is poorly understood, but foliar application of JA during vegetative and flowering stages of soybean development increased the amount of sulfur-containing amino acids in seeds [42]. Increasing expression of genes underlying JA synthesis is a plausible avenue for increased 11S accumulation as a result of increased sulfur-containing amino acid content. Upregulation of JA hormone signaling in high 11S soybeans should be further investigated for a potential positive correlation between JA signaling and 11S accumulation.

*Glyma.06G287800* was upregulated among high 11S soybeans (Table 3). Poly-galacturonase in *M. truncatula* was the top BLASTP identity and the TAIR10 top identity was a Pectin lyase-like superfamily protein (Appendix A). Poly-galacturonases are a family of hydrolases with a roll in cell separation by catalyzing α(1–4) linkages between D-galacturonic acid residues in cell wall pectins [43]. As a result of this role, poly-galacturonases are involved in a range of developmental programs including embryo development, organ abscission and pod dehiscence [44]. An increase in expression of poly-galacturonase in high 11S soybeans may attribute to cell wall loosening during embryo development in anticipation of accumulation of seed storage molecules.

*Glyma.10G141200* was upregulated in high 11S soybeans and was identified to be closely related to the Disproportionating enzyme 1 (DPE1) found in *Phaseolus angularis* (Table 3 and Appendix A). Plastidic DPE1 in rice has a significant role in the starch synthesis pathway by mediating the transfer of maltooligosyl groups from amylose, as well as amylopectin, to amylopectin [45]. The expression of *Glyma.10G141200* in high 11S soybeans, not seen in low 11S soybeans, may be an important factor potentially underlying differences in carbohydrate synthesis and carbon allocation. Knockdown/knockout experiments could shed more light on the specific role expression of *Glyma.10G141200* on starch synthesis and carbon allocation.

*Glyma.18G112500* was upregulated in high 11S soybeans and was identified to be closely related to the Tetratricopeptide repeat (TPR)-like superfamily protein in *Arabidopsis* according to our TAIR10 results (Table 3 and Appendix A). TPR-like proteins are important determents during signal transduction, mediated by plant hormones [46]. Increased expression of *Glyma.18G112500* in high 11S soybeans compared to low 11S soybeans suggests it may have an underlying role in signal transduction and/or hormone signaling which influence 11S content. It would be of interest to conduct protein–protein interaction (PPI) prediction analysis on *Glyma.18G112500* to determine interacting partners.

*Glyma.01G127800* is downregulated in the high 11S soybean lines (Table 3). While yet uncharacterized in *G. max*, the BLASTP and TAIR10 results identify an *Arabidopsis* transducin family protein/WD-40 repeat family protein as the most closely related known sequence (Appendix A). WD40 domains (sometimes called WD-repeat proteins) are prominent features within proteins spanning ~40–60 amino acids, typically terminated by a WD motif. WD40 domains are sites of PPIs, including multicomplex interactions and sometimes act as transient regulators for PPIs [47]. WD40 proteins conserved in *Arabidopsis* have a spectrum of known functions: auxin response, light signaling, meristem maintenance, time of flowering, flowering and seed development. Evidently the transducin family/WD-40 repeat family proteins influence a wide range of processes; the exact function of the protein encoded by *Glyma.01G127800* is not currently understood. Downregulation of *Glyma.01G127800* across high 11S soybeans suggests there is a decrease in WD40-related PPIs, signaling, or regulation of one of these pathways. PPI prediction for *Glyma.01G127800* would be in the interest of directing further investigation into the putative role of this gene on seed content.

Intriguingly, *Glyma.13G077600* and *Glyma.03G054100* both made an appearance on the high TP and the high 11S downregulated shortlists. *Glyma.13G077600* is downregulated in 7/10 high vs. low TP DE analyses and also in 7/10 high vs. low 11S DE analyses (Table 2 and Table 3). *Glyma.13G077600* is uncharacterized in *G. max*, but the top BLASTP hit identified a protein of unknown function in *Arabidopsis* of the DUF538 protein superfamily as the most closely related protein on record (Appendix A). Proteins of the DUF538 superfamily are widely distributed in monocots and dicots, with a conserved recognizable ß-sheet-rich domain called the DUF358 domain [48]. DUF538 proteins have been predicted to play regulatory roles in plants under different stress conditions and may be chlorophyll hydrolyzing enzymes induced by stress response stimuli [48]. The ambiguity of DUF358 proteins in literature makes pinpointing the exact relationship between *Glyma.13G077600* and seed protein difficult. PPI prediction for the protein encoded by *Glyma.13G077600* could be useful in investigating this connection.

*Glyma.03G054100* is downregulated in 5/10 high vs. low TP datasets and also downregulated in 5/10 high 11S datasets (Table 2 and Table 3). *Glyma.03G054100* is predicted in *G. max* to encode a TMV resistance protein N-like isoform X3 (Appendix A). The most closely related protein according to BLASTP results is the TIR-NBS-LRR RCT1-like resistance protein in *Medicago sativa* (Appendix A). Resistance (R) proteins are an important part of the defense system. A specific elicitor such as an avirulence (Avr) protein requires a specific R protein from the host to recognize it to signal the effector triggered immunity (ETI) pathway. Recognizing the Avr protein by specific R proteins leads to a cascade of responses resulting in an immune response by the plant in the form of a localized programmed cell death, through a hypersensitive response [49]. The downregulation of *Glyma.03G054100* in both high TP and high 11S samples may be indicative of reduced ETI pathway signaling in these soybeans. The low 11S and low TP soybeans may induce signaling cascades in response to an environmental stress, which does not affect the high TP and high 11S soybeans in the same way.

Revigo is a tool used to reduce the number of GO terms in a given set to those scored least dispensable and to visualize the ontologies relative to one another in semantic space. Revigo scores GO terms from a given set (GO enrichment analysis) based on frequency, uniqueness and dispensability. Based on these parameters, a relative size for each group of terms is calculated, giving an indication of the relative number of daughter terms which fall under a parent term. GO enrichment for a given set of genes can be evaluated semantically and the GO term population structure and term relationships can be assessed. Using Revigo, we were able to assess GO based on the most indispensable terms across the shortlisted genes. After redundancy reduction, embryo development ending in seed dormancy (GO:0009793) was among the top indispensable downregulated BP GOs in high TP soybeans (Figure 6B), indicating high TP soybeans (lines 8, 9 and 10) are downregulating genes with roles in embryo development and seed dormancy. Upregulated genes in high 11S soybeans (lines 2, 4 and 8) had two key indispensable BP GO terms, lipid metabolic process (GO:0006629) and carbohydrate metabolic process (GO:0005975), signifying a significant evidence of upregulation in expression of genes involved in these two processes in high 11S soybeans (Figure 7A). Among the most important terms upregulated in high 11S soybeans as determined by Revigo include glucose metabolic process (GO:0006006), starch metabolic process (GO:0005982), maltose catabolic process (GO:0000023) and oxylipin metabolic process (GO:0031407) (Figure 7A, Appendix A). These results indicate clear differences in expression of genes related to carbohydrate metabolism and lipid metabolism, particularly in high 11S soybeans.

When comparing the shortlisted data between East and West, the average DE of the majority of the genes is similar, with the exception of a few genes of interest. *Glyma.03G057800* (Rhodanese-like family protein-like protein in *M. truncatula*) and *Glyma.10G092400* (With no lysine (WNK) kinase 3, top *Arabidopsis* homolog) are more highly upregulated in high TP soybeans grown in the East than in the West. Eastern-grown high TP soybeans upregulated *Glyma.03G057800* and *Glyma.10G092400* by an average of log2FC 25.1 and log2FC 24.2, respectively; western-grown high TP soybeans upregulated these genes by an average of log2FC 11.9 and log2FC 10.5, respectively (Appendix A). Rhodaneses are versatile sulfur carrying proteins, which have reactive roles in metabolism and regulatory pathways, including cell cycle control [35]. The observed difference between East and West high TP soybean data could be in part due to the products of *Glyma.03G057800* (regulatory protein) and *Glyma.10G092400* (a kinase) potentially having a relationship. It could be suggested that the high TP soybeans in the East are more tightly regulating cell cycle control and have enhanced intracellular signaling by increasing expression of *Glyma.03G057800* and *Glyma.10G092400*, respectively. However DE analysis between genotypically identical high TP cultivars across East and West growing locations would need to be carried out to confirm this suggestion.

In comparing East vs. West high 11S average DE, the two locations show similar values with the exception of one gene, *Glyma.03G054100*, which is significantly more downregulated in western-grown high 11S soybeans (log2FC −38.9) than eastern-grown high 11S soybeans (log2FC −7.29) (Appendix A). In high TP soybeans *Glyma.03G054100* was downregulated in the East by an average log2FC of −18.58 and in the West by −25.25 (standard deviation of 4.71). *Glyma.03G054100* is predicted in *G. max* to encode a TMV resistance protein N-like isoform X3 and TIR-NBS-LRR RCT1-like resistance protein in *M. sativa* is the most closely related known protein according to BLASTP (Appendix A). Significant downregulation of a resistance gene in high 11S soybeans in the West compared to eastern-grown high 11S soybeans was observed without any differences seen in pathogen pressure. This gene was shortlisted for downregulation in both the high TP and high 11S soybeans, thus, there may be some attribution to a potential role in protein quality.

## 4. Materials and Methods

### 4.1. Lines, Locations, and Planting

The ten selected soybean lines, ranging in seed protein content, were grown in replicated trials at four locations across Canada: Ottawa Ontario (latitude 45.39, longitude −75.72), Morden Manitoba (49.18, X98.08), Brandon Manitoba (49.86, −99.98) and Saskatoon Saskatchewan (52.15, −106.57). Listed from lowest to highest average seed protein content, the lines under investigation are as follows; X5583-1-041-5-5 (line 1); AC Harmony [50] (line 2), AAC Halli (line 3), 90A01 [51] (line 4), Maple Amber (line 5), OT13-08 (line 6), OT14-03 (line 7), AAC Springfield (line 8), Jari (line 9) and AC Proteus [52] (line 10). These lines were selected as a representation of the spectrum of seed protein and oil in Canadian soybean agriculture. This spectrum of phenotypic information offered multi-cultivar information on high and low TP and/or 11S soybeans to dilute any genotypic bias. Planting was done in mid-end of May each year. Trial arrangement was carried out in 4 × 5 rectangular lattices and each had four replicates. Plot planting density was at a rate of 50 seeds m^2^ and best management practices were taken by each site. For more details on lines, see [53].

### 4.2. Sampling and RNA Extraction

Triplicate leaf tissue samples from R5 stage [54] soybeans for each of the 10 lines in each location were collected annually since 2018 and subjected to RNA-seq creating a large, high-quality data set. Tissue samples were crushed in liquid nitrogen using RNase-treated mortar and pestle. RNA extractions using SPLIT Total mRNA Extraction Kit (Lexogen, Vienna, Austria) was performed on approximately 200 mg of crushed leaf tissue from each sample according to the manufacturer’s instructions. RNA quality was initially tested using a NanoDrop™ 2000 Spectrophotometer (Thermo Fisher Scientific, Waltham, MA, USA), followed by a number of quality analysis checkpoints including agarose gel electrophoresis, TapeStation 4200 RNA ScreenTape (Agilent, Santa Clara, CA, USA) and 2100 Bioanalyzer (Agilent, Santa Clara, CA, USA) at Génome Québec (Montréal, QC, Canada) and the Ottawa Research and Development Centre (Ottawa, ON, Canada). RNA with RIN values of at least 4.5 and a Q30 score of at least 36 were used for library preparation. Spike-in RNA variants (SIRVs) (Lexogen, Vienna, Austria) were integrated within the RNA samples as controls to monitor and compare key parameters (such as sensitivity and quantification). For this work the E0 SIRV mix is used, which contains 69 different isoform variants with known sequences added at the same molar concentration.

### 4.3. RNA-seq Library Preparation, Alignment, Read Mapping, Read Counting, and DE

cDNA library preparation with paired-end sequencing was carried out using the Illumina HiSeq 4000 platform (Illumina, San Diego, CA, USA). RNA-seq analysis was carried out by Génome Québec and the preliminary DE analysis was performed by the department of bioinformatics at McGill University (Montréal, QC, Canada). Each read was sequenced and exported as a FASTA file for downstream RNA-seq quality control (QC) and analysis. dupRadar [55] (v3.16, Biberach an der Riß, Germany; Bioconductor, R) was used for duplication rate QC for the RNA-seq data.

Read normalization at the individual-sample level was performed using edgeR [56] (v3.16, Parkville, Victoria, Australia) and normalized data was subjected to exploratory data analysis (EDA) using R. PCA outputs for RNA-seq normalized read data were used to construct Figure 2 and Figure 3 in order to compare variance between transcript data. Each sample on the PCA plots is identified by its line number. The PCA plots do not directly represent the seed content variability data, but in some instances TP/11S content is responsible for the first PC (e.g., Ottawa 2018 and Saskatoon 2019 for the TP data).

QualiMap [57] (v2.2.1, Berlin, Germany) is a program which runs on an independent platform and was used as a QC step for the alignment of the sequencing data and features (genes or transcripts). Preseq [58] (v3.1.1, Los Angeles, CA, USA; Bioconductor, R) was used to predict the number of distinct reads from a sequencing library (in this case, RNA-seq). RSeQC [59] (v4.0.0, Nanjing, China; Bioconductor, R) was used to comprehensively evaluate the RNA-seq read data. RSeQC calculates the semantic read distribution of a sample, the inner distance between two reads, presence of exact read duplications, junction saturation, etc.

GenPipes is the main in-house framework of the Canadian Centre for Computational Genomics used to perform major processing steps [60]. Genpipes is the pipeline used to automatically run all the pre-processing steps mentioned below. Adaptor sequences and low-quality score containing bases (Phred score < 30) were trimmed from reads using Trimmomatic [61] (v0.36, Jülich, Germany). The resulting reads were aligned to the soybean genome (Glycine_max_v2.1, INSDC Assembly GCA_000004515.4, Jul 2018), using STAR [62] (v2.7.7a, Menlo Park, CA, USA) under the following command—runMode alignReads, after generating out own index files from the aforementioned genome. Read counts are obtained using HTSeq [63] (v0.12.3, Heidelberg, Germany) using the following options: “-m intersection-nonempty”. 

The R package DESeq2 [64] (v3.16, Heidelberg, Germany) was used to identify differences in expression levels between the groups using negative Binomial GLM fitting and Wald statistics: nbinomWaldTest, similarly “ashr” [65] was used to shrink log2 fold changes in gene expression data. Genes were considered to be differentially expressed when they had a adjusted *p*-value < 0.05 as well as a log2 Fold change > |2| for shortlist considerations. For TP DE analyses, “low TP” samples serve as the “control” expression values; the expression of genes in low TP soybeans is being used to compare the expression of genes in high TP soybeans. With this, it is important to note when discussing log2FC differences in expression that the difference is seen in the high TP group. Upregulated genes would therefore be upregulated in high TP soybeans; downregulated genes are downregulated in high TP soybeans. Analyses for 11S protein content and the 11S:7S ratio was analyzed following the same fashion. Location was always controlled for each DE dataset, i.e., high protein soybeans grown in Ottawa were only compared with low protein soybeans grown in Ottawa.

### 4.4. Determination of Soybean Sample Seed Content

Seeds from individual plots from each soybean line-location-year trial were analyzed for the content of protein, oil, 11S, 7S and 11S:7S ratio at the central grain quality lab at Agriculture and Agri-Food Canada (AAFC) Ottawa Research and Development Centre (ORDC). Protein and oil content were determined using a Foss grain analyzer (Infratec 1241, Foss, Eden Prairie, MN, USA). Protein quality samples were milled in a ball mill before defatting via Accelerated Solvent Extraction Machine (Thermo Scientific Dionex ASE 350, Waltham, MA, USA). Protein separation was carried out using a Bioanalyzer (Agilent 2100 Bioanalyzer, Santa Clara, CA, USA) using a Protein 230 kit for sample preparation. Electropherogram analysis was used to assess 11S and 7S subunits and subsequently used to calculate the 11S:7S ratio. See [53] for a detailed description. Three samples with the highest and three samples with the lowest average values across all years (2018–2020) were designated “high” and “low”, respectively.

### 4.5. Candidate Gene Selection

As stated above, the DE datasets were trimmed at an adjusted *p*-value < 0.05 and a log2FC of at least 2. The total number of DE genes following trimming for each dataset is listed in Appendix A. The average number of remaining genes following trimming was 101 genes, thus the top and bottom 15% of the average number of genes with a log2FC of at least 2 was selected for cross examination (15 genes). From each DE dataset (10 TP datasets or 10 11S datasets), the top 15 upregulated genes (150 genes) and 15 most downregulated genes (150 genes) were extracted for a total of 300 genes differentially expressed in high TP soybeans and 300 genes differentially expressed in high 11S soybeans. From each of the resultant lists, the most frequently occurring genes were selected for candidacy. Genes that were found to be within the top 15 DE genes in at least 5/10 DE datasets were shortlisted. Table 4 lists all shortlisted genes and their corresponding NCBI gene identities.

### 4.6. Gene Ontology and Pathway Analysis

Shortlisted genes were assessed for GO to better understand the key BPs and MFs of genes in the curated candidate lists. Resultant gene lists, both up- and downregulated, were run independently through the SoyBase GO Term Enrichment Tool [22] (https://www.soybase.org/ (accessed on 17 May 2022)) to curate a list of GO terms representative of each gene list.

To search specifically for protein- and lipid-related genes, GO terms were identified using QuickGO (https://www.ebi.ac.uk/QuickGO/ (accessed on 17 May 2022)) [67] and supported by findings from a recent study on identifying soybean genes related to seed protein content [68]. Search keyword terms related to lipids include those with descriptions including “fat”, “glyce” and “lip” to encompass descriptive terms related to lipid prosses. Terms related to seed storage protein were searched based on the GO term for cupins, vicilins and globulins; “nutrient reservoir”. Because “nutrient reservoir” could include genes other than seed storage proteins, genes associated with this term were confirmed using the Nation Centre for Biotechnology Information (NCBI) for their annotation as a seed storage protein-related gene. Each curated GO list was then run through Revigo [23,24] (http://revigo.irb.hr/ (accessed on 17 May 2022)) at a threshold of 0.5 (50%) using *Arabidopsis thaliana* as the closest reference species.

### 4.7. Expression Profile Matrices and Heatmaps

Expression profiles were normalized across each dataset and exported as a matrix by edgeR [56]. edgeR uses a RNA-seq-specific normalization function for expression data for all samples in an expression matrix (produced by salmon). Genes of interest to be explored via heatmaps were selected based on the above criteria for curation of candidate gene shortlists.

Expression profiles can be thought of as a library of all genes and their relative expression in each individual sample. Heatmaps are a convenient way to compare the expression of a set of genes across all individual samples while simultaneously looking at the expression data globally. A matrix of all relative expression data of the genes of interest across all individual samples is read into the data processing tool, Heatmapper [69,70] (http://www.heatmapper.ca/ accessed on 14 June 2022), including all normalized expression data regardless of log2FC DE values for comparative expression purposes. Genes and their corresponding expression data are arranged according to the specified hierarchical gene clustering method (average linkage) and distance measurements (Euclidean). The correlation coefficient (distance measurement) between each pair of variables (columns, or individual samples) is calculated and a matrix of pairwise coefficients is created [69]. The row Z-score is used to scale the normalized expression data for enhanced visualization of trends in heatmaps and is calculated by (gene expression value in sample of interest)—(mean expression across all samples)/(standard deviation) [71].

## 5. Conclusions

In this work we sought to find DE genes between high and low TP soybean lines and between high and low 11S protein soybean lines from samples grown in four locations across Canada’s growing regions over 3 years. We identified shortlists of upregulated and downregulated genes in high TP and high 11S soybeans which may be of significance to soybean seed protein breeding programs. Ontologies of these genes include those within embryonic development, lipid metabolic pathways and carbohydrate metabolic pathways which may hold the key to the difference in seed quality between our selected lines. Within these shortlisted genes, a handful of key genes were found to be disproportionately DE between East and West growing locations. This suggests that these genes are likely to underlie the molecular mechanism responsible for the long withstanding observation of different seed protein and oil content, influenced by the different environmental factors at play.

## Figures and Tables

**Figure 1 ijms-24-00222-f001:**
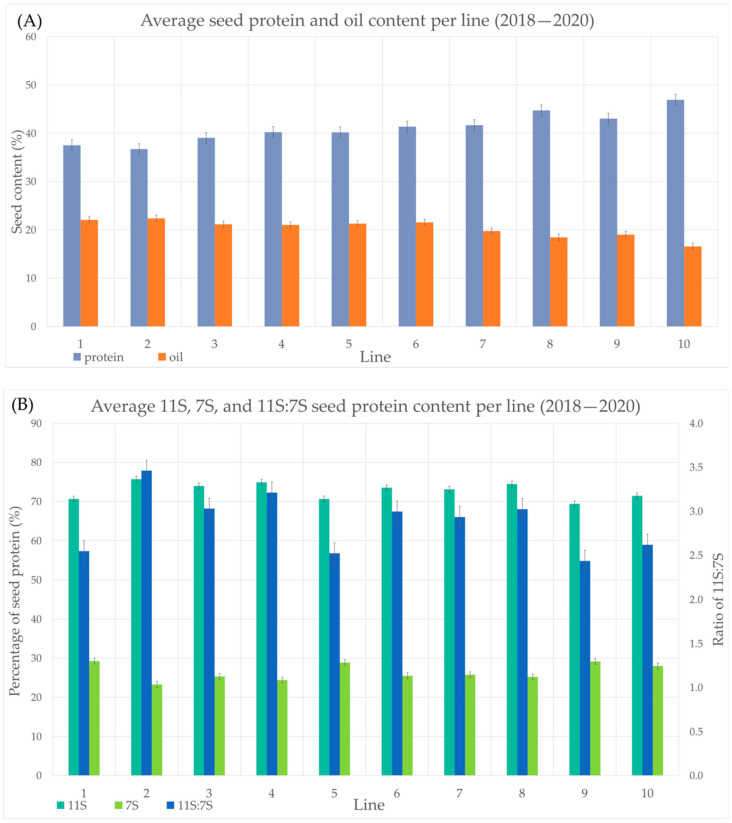
Seed composition across each variety from 2018–2020. (**A**) Average TP and seed oil content as a percentage of the seed weight. (**B**) Average 11S and 7S protein content as a percentage of total seed protein (primary y axis) and the average 11S:7S ratio (secondary y axis). Error bars indicate the least squares difference per series.

**Figure 2 ijms-24-00222-f002:**
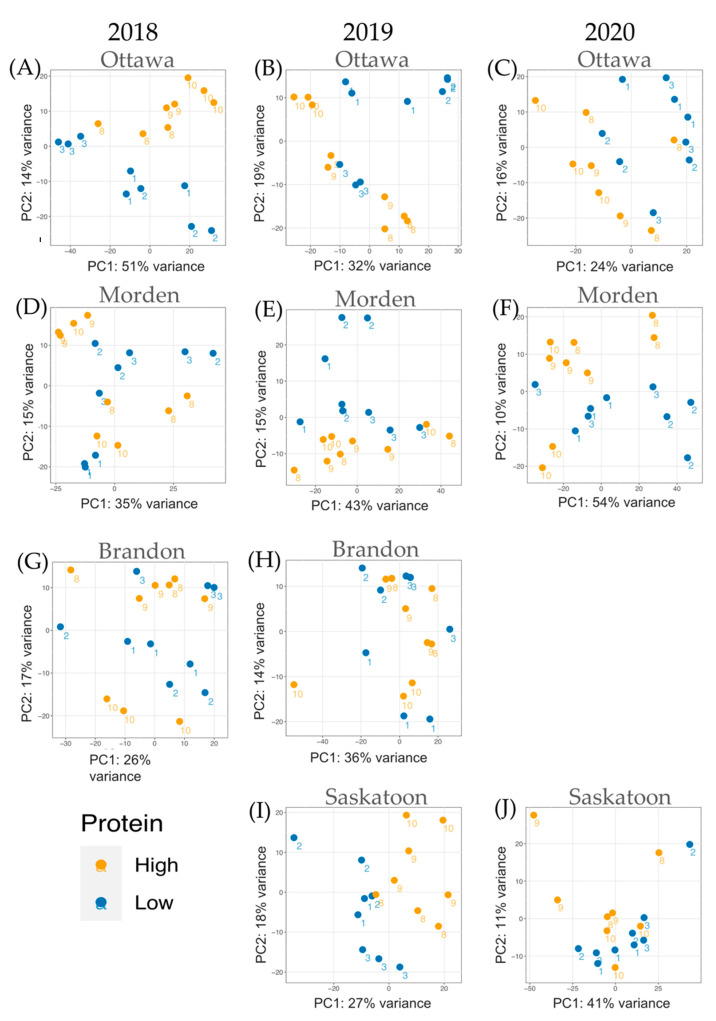
PC analysis of normalized RNA-seq expression data for samples included in each individual DE analysis across the four locations and three years. Each TP year-location dataset is represented: (**A**) Ottawa 2018, (**B**) Ottawa 2019, (**C**) Ottawa 2020, (**D**) Morden 2018, (**E**) Morden 2019, (**F**) Morden 2020, (**G**) Brandon 2018, (**H**) Brandon 2019, (**I**) Saskatoon 2019, (**J**) Saskatoon 2020. Orange datapoints represent high TP samples, blue datapoints represent low TP samples. No data collected from Saskatoon 2018 and Brandon 2020. The line number (1, 2, 3, 8, 9, 10) for each corresponding datapoint is indicated.

**Figure 3 ijms-24-00222-f003:**
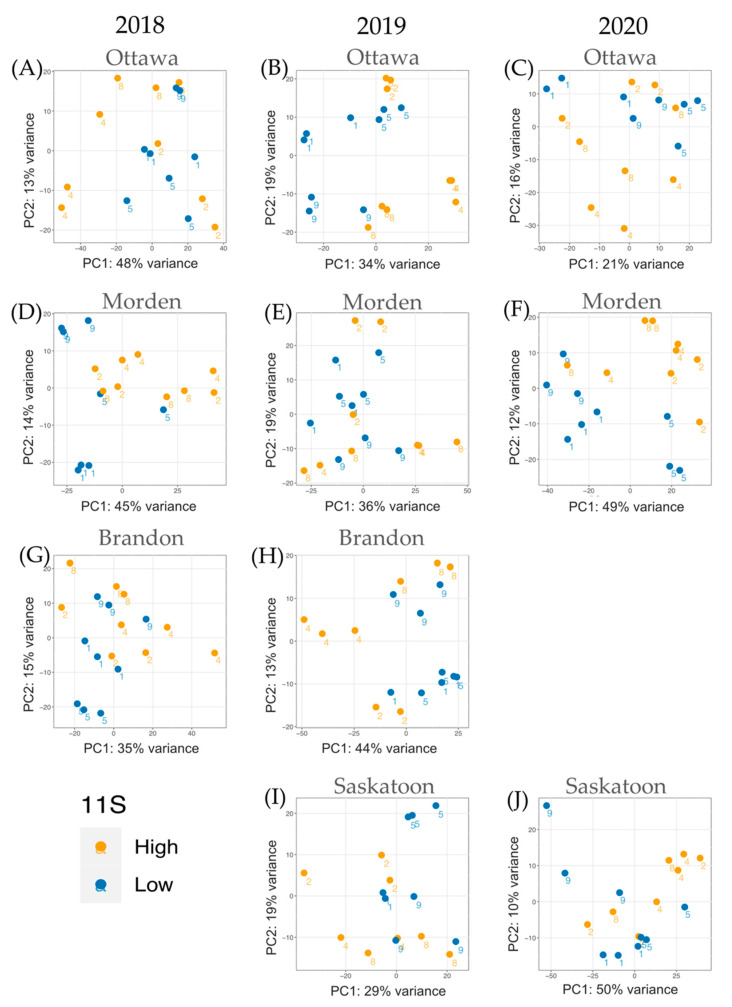
PC analysis of normalized RNA-seq expression data for samples included in each individual DE analysis across the four locations and three years. Each 11S year-location dataset is represented: (**A**) Ottawa 2018, (**B**) Ottawa 2019, (**C**) Ottawa 2020, (**D**) Morden 2018, (**E**) Morden 2019, (**F**) Morden 2020, (**G**) Brandon 2018, (**H**) Brandon 2019, (**I**) Saskatoon 2019, (**J**) Saskatoon 2020. Orange datapoints represent high 11S samples, blue datapoints represent low 11S samples. No data for Saskatoon 2018 and Brandon 2020. The line number (1, 2, 4, 5, 8, 9) for each corresponding datapoint is indicated.

**Figure 4 ijms-24-00222-f004:**
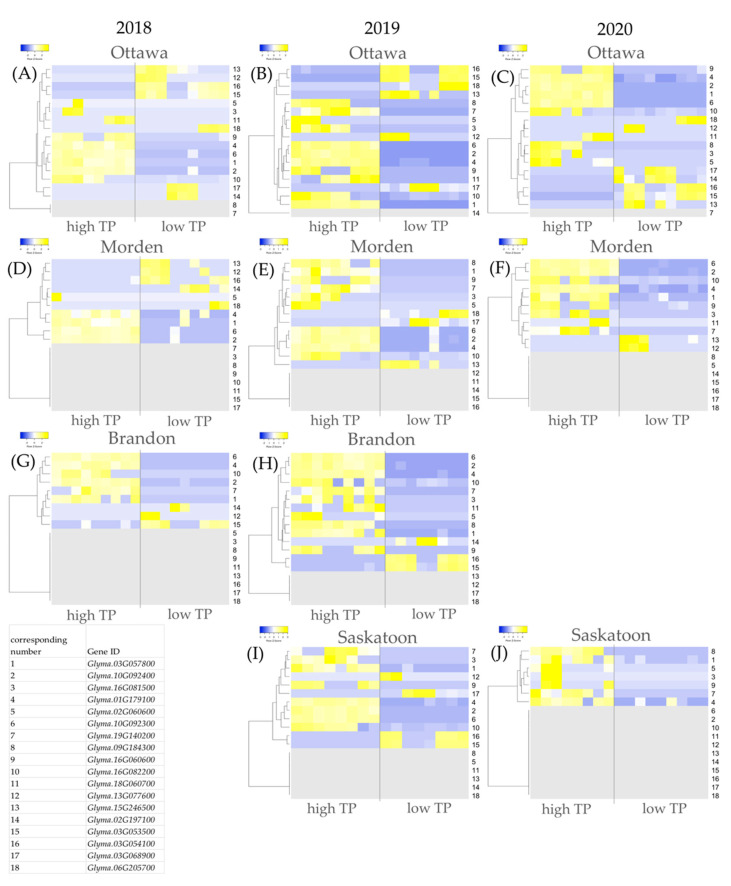
Expression heatmaps for shortlisted genes differentially expressed in high TP soybeans (lines 8, 9, 10 vs. lines 1, 2, 3) for each year-location analysis. Each TP year-location dataset is represented: (**A**) Ottawa 2018, (**B**) Ottawa 2019, (**C**) Ottawa 2020, (**D**) Morden 2018, (**E**) Morden 2019, (**F**) Morden 2020, (**G**) Brandon 2018, (**H**) Brandon 2019, (**I**) Saskatoon 2019, (**J**) Saskatoon 2020. Upregulation is indicated by shades of yellow, downregulation is indicated by shades of blue. Grey indicates that a gene was not differentially expressed between high and low TP soybeans at an adjusted *p*-value < 0.05. The legend at the bottom left provides the gene identities and their correspondence with the numbers on the right y-axis of the heatmaps.

**Figure 5 ijms-24-00222-f005:**
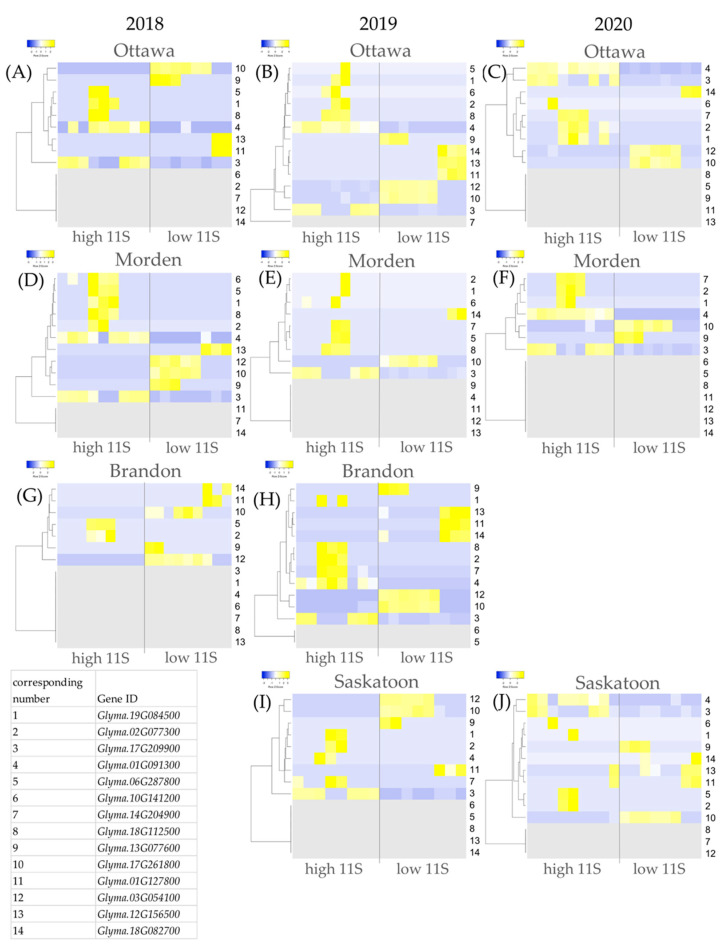
Expression heatmaps for shortlisted genes differentially expressed in high 11S soybeans (lines 2, 4, 8 vs. lines 1, 5, 9) for each year-location analysis. Each 11S year-location dataset is represented: (**A**) Ottawa 2018, (**B**) Ottawa 2019, (**C**) Ottawa 2020, (**D**) Morden 2018, (**E**) Morden 2019, (**F**) Morden 2020, (**G**) Brandon 2018, (**H**) Brandon 2019, (**I**) Saskatoon 2019, (**J**) Saskatoon 2020. Upregulation is indicated by shades of yellow, downregulation is indicated by shades of blue. Grey indicates that a gene was not differentially expressed between high and low 11S soybeans at an adjusted *p*-value < 0.05. The legend at the bottom left provides the gene identities and their correspond with the numbers on the right y-axis of the heatmaps.

**Figure 6 ijms-24-00222-f006:**
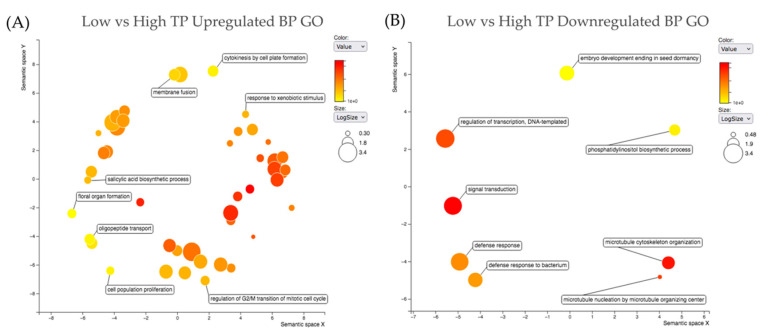
Revigo plots summarizing the relationships between the most indispensable biological process (BP) gene ontologies (GOs) for upregulated (**A**) and downregulated (**B**) shortlist genes across the high vs. low TP DE analyses. Circle size represents logSize value, higher logSize values indicate a strong presence of a term and/or its daughter terms; more general terms have larger bubbles. Color represents significance of a term among the query set of GOs.

**Figure 7 ijms-24-00222-f007:**
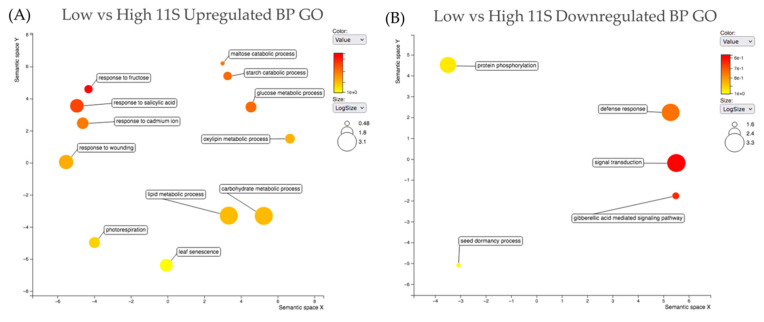
Revigo plots summarizing the relationships between the most indispensable BP GO terms for the upregulated (**A**) and downregulated (**B**) shortlist genes across the high vs. low 11S DE analyses. Circle size represents logSize value, higher logSize values indicate a strong presence of a term and/or its daughter terms; more general terms have larger bubbles. Color represents significance of a term among the query set of GOs.

**Table 1 ijms-24-00222-t001:** The average seed protein and oil content (as a percentage of the seed weight) for the lines selected for high vs. low TP DE (lines 1, 2, 3 vs. lines 8, 9, 10). The average 11S and 7S content (as a percentage of total seed protein content) and the average ratio of 11S:7S from high 11S lines selected for high vs. low 11S DE analysis (lines 2, 4, 8 vs. lines 1, 5, 9). The *p*-values represent the statistical difference between high and low soybean groups for each attribute via two-tailed *t*-tests.

High and Low TP Lines for DE				High and Low 11S Lines for DE				
	**Line**	**TP**	**oil**		**Line**	**11S**	**7S**	**11S:7S**
Low TP	1	37.5	22.1	Low 11S	1	68.9	30.8	2.38
	2	36.7	22.4		5	69.0	30.2	2.37
	3	38.9	21.2		9	68.0	30.8	2.28
High TP	8	44.6	18.5	High 11S	2	74.1	24.9	3.19
	9	43.0	19.0		4	72.5	25.8	2.97
	10	46.9	16.5		8	72.4	27.3	2.78
*p*-value		0.010	0.018	*p*-value		0.005	0.016	0.027

**Table 2 ijms-24-00222-t002:** The log2FC DE between low TP soybeans (lines 1, 2, 3) and high TP soybeans (lines 8, 9, 10) from 2018–2020 over 4 different locations. The top 15 most upregulated and 15 most downregulated genes from each DE analysis were selected. Genes DE in at least 5 of 10 datasets are shortlisted here including the log2FC DE in the high seed protein samples. The 10 DE datasets are identified by their Year-Location shorthand; as an example, 18.B indicates DE of high vs. low 11S soybeans in 2018 from Brandon. O, Ottawa; M, Morden; B, Brandon; S, Saskatoon.

log2foldchange	−25	−20	−15	−10	−5	0	5	10	15	20	25
	High Total Protein vs. Low Total Protein										
Gene_id	18.B	18.M	18.O	19.B	19.M	19.O	19.S	20.M	20.O	20.S	Total
*Glyma.03G057800*	18.6		22.0		24.3	27.5	5.93	5.14	25.7	5.51	8
*Glyma.10G092400*	8.62	5.08		8.83	5.36	22.9	24.2	10.8	25.5		8
*Glyma.16G081500*			23.8	28.5	29.6	31.0	29.5	31.8	30.8	30.2	8
*Glyma.01G179100*	24.7		24.5	9.37			7.41	7.05	5.65	2.53	7
*Glyma.02G060600*		26.7	27.6	34.3	35.7	35.9			35.1	34.8	7
*Glyma.10G092300*	25.6	4.84		25.0		22.5	25.0	12.4	27.1		7
*Glyma.19G140200*	18.4			21.2	21.5	22.8	22.1	21.2		21.5	7
*Glyma.09G184300*				22.6	23.6	23.2			22.8	23.5	5
*Glyma.16G060600*				23.2	23.6	23.8	24.3			23.0	5
*Glyma.16G082200*	18.9		7.32		6.50		5.06	5.00			5
*Glyma.18G060700*			20.8	21.5		23.5		21.1	20.4		5
*Glyma.13G077600*	−13.4	−35.2	−34.7			−35.4	−32.6	−35.1	−31.7		7
*Glyma.15G246500*		−19.2	−20.8		−26.0	−26.3		−24.2	−22.5		6
*Glyma.02G197100*	−11.3	−33.3	−36.7	−42.7					−38.1		5
*Glyma.03G053500*			−22.4	−9.52		−25.8	−25.9		−24.1		5
*Glyma.03G054100*			−23.4	−25.9		−10.2	−24.6		−22.2		5
*Glyma.03G068900*			−23.9		−24.6	−24.3	−24.0		−23.2		5
*Glyma.06G205700*		−18.1	−20.7		−21.8	−23.9			−19.6		5

**Table 3 ijms-24-00222-t003:** log2FC DE between low 11S soybeans (lines 1, 5, 9) and high 11S soybeans (lines 2, 4, 8) from 2018–2020 over 4 different locations. The top 15 most upregulated and 15 most downregulated genes from each DE analysis were selected. Genes are DE in at least 5 of 10 datasets are listed here including the log2FC DE in the high 11S samples. DE datasets are identified by their year-location shorthand; as an example, 18.B indicates DE of high vs. low 11S soybeans in 2018 from Brandon. O, Ottawa; M, Morden; B, Brandon; S, Saskatoon.

log2foldchange	−25	−20	−15	−10	−5	0	5	10	15	20	25
	High 11S vs. Low 11S										
Gene_id	18.B	18.M	18.O	19.B	19.M	19.O	19.S	20.M	20.O	20.S	Total
*Glyma.19G084500*		24.7	22.3	25.3	22.9	27.2	21.5	27.5	27.3	24.7	9
*Glyma.02G077300*	16.9			22.0	16.1	21.8	19.0	22.0	22.2	22.2	8
*Glyma.17G209900*			6.31	7.44	6.16	7.63	5.89	8.24	10.6	3.45	8
*Glyma.01G091300*				7.43			4.93	7.25	9.21	6.86	5
*Glyma.06G287800*	18.6		13.8		22.1	18.7				21.2	5
*Glyma.10G141200*		18.9			20.7	17.0			11.5	4.50	5
*Glyma.14G204900*				23.6	22.4		22.3	22.7	22.2		5
*Glyma.18G112500*		20.3	20.8	23.8	24.0	24.8					5
*Glyma.13G077600*		−33.2	−33.5	−30.8		−33.5	−30.6	−32.8		−29.6	7
*Glyma.17G261800*	−15.0	−38.6	−40.0		−46.0		−44.9		−42.4	−42.0	7
*Glyma.01G127800*			−40.2	−43.2		−45.2	−43.5			−41.6	5
*Glyma.03G054100*	−18.6	−42.7		−46.1		−7.29	−48.3				5
*Glyma.12G156500*		−20.6	−20.9	−21.3		−22.6				−20.8	5
*Glyma.18G082700*				−39.8	−42.2	−43.9			−41.0	−38.7	5

**Table 4 ijms-24-00222-t004:** All short-listed *G. max* gene IDs and corresponding NCBI Gene ID (https://www.ncbi.nlm.nih.gov/ accessed on 3 October 2022) [66].

Gene ID (Wm82.a2)	NCBI Gene ID
*Glyma.01G091300*	NA
*Glyma.01G127800*	NA
*Glyma.01G179100*	102669100
*Glyma.02G060600*	100808728
*Glyma.02G077300*	NA
*Glyma.02G197100*	NA
*Glyma.03G053500*	NA
*Glyma.03G054100*	NA
*Glyma.03G057800*	100780425
*Glyma.03G068900*	100527900
*Glyma.06G205700*	NA
*Glyma.06G287800*	NA
*Glyma.09G184300*	106794632
*Glyma.10G092300*	NA
*Glyma.10G092400*	100805392
*Glyma.10G141200*	NA
*Glyma.12G156500*	NA
*Glyma.13G077600*	NA
*Glyma.14G204900*	NA
*Glyma.15G246500*	100812621
*Glyma.16G060600*	NA
*Glyma.16G081500*	NA
*Glyma.16G082200*	100791376
*Glyma.17G209900*	100817099
*Glyma.17G261800*	100794722
*Glyma.18G060700*	NA
*Glyma.18G082700*	NA
*Glyma.18G112500*	100787722
*Glyma.19G084500*	NA
*Glyma.19G140200*	NA

NA = Not applicable; no corresponding NCBI gene ID.

## Data Availability

Not applicable.

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
