# Peer review of "A Multi-Year, Multi-Cultivar Approach to Differential Expression Analysis of High- and Low-Protein Soybean (*Glycine max*)"

_ijms, 2022, doi:10.3390/ijms24010222_

Round 1

Reviewer 1 Report

Soybean is one of the most important crops in the world. Serving as the main source of protein and oil, high protein or oil contents of soybean are goals of modern cultivar breeding. Authors of this study planted multiple cultivars of soybean across Canada for three consecutive years and collected data for downstream analyses, including physiological and differential expressed genes (DEGs) analyses. However, the entire experimental design has great flaws that I cannot accept the results and conclusions. Here are my comments:

1. Ten cultivars were planted across Canada. Firstly, the authors didn't explain why these lines were chosen. Secondly, based on the results of the seed content determination, three cultivars with either high or low contents of TP and 11S were selected for DEG analysis. However, I doubt if these lines are significantly different on seed contents as shown in Figure 1, lower contents has larger bars. The authors failed to provide results of the statistic comparison among these lines. Indeed, in Figure 2 and 3, these lines couldn't be consistently designated into separated groups based on TP or oil contents. 

2. Only 15 top up- and down-regulated genes were selected for downstream analysis. This is extremely unacceptable. This is not the way we conduct DEG analysis. The better way is to set up a certain threshold for significantly DEG and keep all the genes passing the filter. Besides, the authors used DESeq for DEG identification, which is outdated for a while and has a more successful sequel. The authors even didn't use the statistic results such as q value imbedded in this package. 

3. In this research, the authors only included 15 for up- or down-regulated genes and even shortened the list by requesting partial consistency among cultivars. This makes the enriched GO terms unreliable as many other DEGs were excluded from the analysis. 

4. The methods of data analysis are far from repeatable. The authors only listed the names of software they used and shortly described their functions. However, no parameters were provided. In addition, they used two pieces of software for read mapping, Subread and STAR. This is confusing. 

5. Though the language is good, the entire manuscript is full of description but not conclusions or interpretations. This even happens in the discussion part. They also cited too many references. This is not appropriate as only necessary references should be cited.

Author Response

The authors sincerely thank Reviewer 1 for reviewing our manuscript and offering detailed comments on areas in need of improvement.

To address comment 1, the ten lines were selected to represent the spectrum of seed protein content in Canadian soybean agriculture. By selecting multiple cultivars from a pool of samples along the spectrum of protein content we avoid genotypic bias (ie. a single QTL which may account for the majority of the difference in protein). We had amended the manuscript to explain this in the manuscript. Comment 1 brings into question the statistical analysis of differences in seed content. The error bars in Figure 1 represent the standard deviation within each phenotype series (protein, oil, 11S, 7S, 11S:7S) including all 10 genotypes which dilutes the significance of the difference (largely due to the samples that fall in the middle ranges of TP and 11S). The standard deviation has been added to Supplementary Table 1 to give readers a finite value to gain perspective on the phenotypic data across the 10 lines. Further, Reviewer 1 points to the statistical phenotypic differences between the lines used for DE analysis; we appreciate the attention being drawn to this detail as it will certainly be important to include. Two-tailed t-test information for high/low TP seed content data (p-value 0.01) as well as high/low 11S seed content data (p-value 0.005) has been added to Table 1, as well as in text descriptions of the data. We agree Figures 2 and 3 lend some ambiguity, the data is the result of experimental field data which tends to be imperfect data to work with. We would like to clarify that the PCA plots in Figures 2 and 3 are constructed based on the raw RNAseq read data, representing the variability between each sample from a transcriptional level. Each sample on the PCA plots is identified by it’s line number (determined by protein/11S content). The PCA plots do not directly represent the seed content variability data, however in some instances protein/11S content is responsible for the first PC. As an example focusing on PC1 between two datasets: Saskatoon protein 2019 has a much better segregation on PC1 than Brandon 2019, thus PC1 for Saskatoon 2019 explains that around 30% of variance can be attributed to protein content but not so much for Brandon 2019 which most likely PC1 is not explained by protein content. This has been now clarified in the manuscript. To further the cumulation of all of the above, we felt the phenotypic data is a good representation of the extreme ends of the phenotypic spectrums for which we are interested (11S and TP). This is part of the reason we opted for a multi-year study, as annual variance between samples was inevitably with field data.

Reviewer 1’s second comment brings into question the threshold cutoffs used to narrow down candidate genes. The cutoff values have been clarified in the methods section. Following DE data trimming, the remaining numbers of DE genes from each dataset has been listed in the newly added Supplementary Table 5. It is with this additional information we hope the perhaps unorthodox selection of 15 top and bottom genes: on average, 101 genes remained after trimming (average across TP and 11S data). The decision to select 15 topmost and bottommost genes is a representation of the top and bottom 15% of DE genes, respectively. This has now been clarified in the manuscript. Comment 2 also points out the use of DESeq; this is a typographical error and has been amended to reflect the correct program used, DESeq2, and the corresponding reference has been updated. The q value embedded in the DESeq2 package (also known as the adjusted p-value) was one of the primary attributes used to trim the data, thus we hope Reviewer 1 can rest assured we have factored this into our study.

In comment 3 from Reviewer 1, the genes included in the GO enrichment analysis are brought into question. The GO enrichment was carried out using all shortlisted genes as a means of gaining perspective on key functions and processes differing between soybean groups. The shortlist of DE genes from the TP data had 76 GOs across the 18 genes, and the shortlist DE genes from the 11S data had 27 GOs across 14 genes. We felt the number of GOs spanning the shortlisted genes was represented best using GO enrichment and Revigo, giving a snapshot of the ongoing functions and processes within the most up- and downregulated genes only. We acknowledge that the results only summarize the shortlisted genes, and not the relationships between them; because of this, the KEGG pathway analysis between shortlisted genes has been removed from the manuscript.

Comment 4 from Reviewer 1 suggests clarification of software packages used. We thank Reviewer 1 for calling our attention to this typographical error, which has now been amended to reflect the correct mapping package. Significant clarifications to the methods have been made, including parameters for the software packages used as well as trimming parameters. We thank Reviewer 1 for drawing our attention to these areas in need of clarification in order to most appropriately communicate the methods.

In Comment 5, Reviewer 1 points to the overuse of referencing material. We agree the list of references is extensive; references have been streamlined to include only those directly relevant to the presented ideas. Some references were added which were essential to offering more conclusive context to our results. Reviewer 1 also notes in Comment 5 that the manuscript needs more interpretations and conclusions with less focus on descriptions. Adjustments and clarifications have been made throughout the manuscript to be more explicit in interpretations and conclusions and unnecessary discussion was removed.

Reviewer 2 Report

Authors searched for soybean genes that are differentially expressed between high and low total protein content as well high and low 11S protein content. Authors selected genes that may be responsible for changes in lipid and carbohydrate metabolism in these subgroups. Study is well planned and performed. Results support conclusions. Some minor improvements as addition of volume and concentration of cDNA libraries should be included. Also a typographical error- in line 450 should be BLASTP not BLASP.   

Author Response

The authors sincerely thank Reviewer 2 for taking the time to review our study and providing thoughtful comments on our experimental design, results and conclusions. The typographical error has been fixed; we appreciate Reviewer 2’s keen eye here. We graciously thank Reviewer 2 for seeing the value in the information we have found in our study.

Round 2

Reviewer 1 Report

Thanks for the authors' efforts to improve this manuscript. I still have some concerns regarding these comments and replies. 

1. The authors have added error bars to the figure, but the figure is still not readable. Next, though the authors admitted that variations existed in the field data, they did not consider these variations in the comparison by using simple means for t-test. In this case, I suggest a pairwise ANOVA or similar comparisons between cultivars for determining high and low groups. 

2. The authors defined ~101 DE genes, and this is not a large number. I still do not understand why only 30 (top and bottom 15% according to the authors) were chosen for downstream analysis. I suggest that the authors may re-analyze the whole DE gene lists.

3. To replying to the comment 4, the authors claimed that the original issue is caused by typo. However, the authors almost rewrote the whole subsection and many different pieces of software were mentioned in this version. It is hard for me to believe the claim. 

4. The authors claimed they had shorted the reference list. However, there were still 146 reference in this version (148 in version 1). I suggest that they include only those very close to current contents and shrink the list to around 70. For the discussion, the authors made short interpretations and perspectives after the descriptions. However, there is few modifications made to the descriptions. Please streamline this section by removing unnecessary descriptions of previous findings.

Author Response

We sincerely thank Reviewer 1 for providing constructive feedback on the revised manuscript. We appreciate the comments provided to evolve our manuscript into it’s strongest version. We have made significant changes to the manuscript, in particular the Discussion section, which we feel has drastically improved the presented work.

In comment 1, Reviewer 1 points to the clarity (or lack thereof) of statistical significance spanning the seed protein content values for the 10 genotypes selected. To address this, we have changed the standard deviation error bars to be LSD error bars in Figure 1 to allow the reader to see significant differences between genotypes. Figure 1 has also been enlarged for better readability. We hope with the addition of t-test statistical values and reformatting Figure 1 we have satisfied Reviewer 1’s comment.

Comment 2, Reviewer 1 points to the strict criteria used in selection of shortlisted genes. We have selected the very topmost DE genes for consideration in this analysis as this is a study which spans multiple years, locations, and cultivars for an expansive and unique perspective on seed protein content differences. Based on the remaining genes following trimming at a log2FC of 2, taking only the top/bottom 15% (total 30% of all genes DE by at least log2FC 2) felt appropriate for our analysis as a representation of the top DE genes. The entire dataset (including all log2FC values) is still considered in the results in both text descriptions and heatmaps. We felt it was important to share our findings with other researchers who may find value in this work.

In response to comment 3, the Reviewer 1 points to the fact that the software section of the methods changed substantially between versions. Following the first round of peer-review, while the typo was corrected, we wanted to clearly satisfy the comments regarding the lack of repeatability due to the lack of parameters. The data analyst on our team (G.Z.) clarified the correct software packages and parameters used during the revision. We contend that we have written our manuscript and responded to all the comments as ethically and honestly as possible and hope we have satisfied Reviewer 1’s comments regarding the software methods and parameters.

In comment 4, Reviewer 1 suggests a significant reduction in reference material is important for improvement of the manuscript. We have taken Reviewer 1’s advice on removing unnecessary descriptions of previous findings, keeping only those most important for discussion context. We have streamlined the entire discussion section while interpretations and perspectives on the findings have been improved. We feel the discussion stands much stronger because of the suggestions made by Reviewer 1, and sincerely thank the Reviewer for the advice. We have removed redundant and unnecessary references and the final reference list sits at 70 in total. Table 4 was a source of many unnecessary supportive references and has been amended to exclude the “Ref” column, as all appropriate references are now included in the discussion. We sincerely hope the updated version of the manuscript reflects our honest attempt at reassessing the presented work. We feel the manuscript has evolved by leaps and bounds with the help of the Reviewers’ diligent evaluation.

Round 3

Reviewer 1 Report

I am satisfied with the responses and modifications the authors made to improve this manuscript, except for the figure layout. I still want the authors can enlarge the font size of labels in the figures, to at least 7. For table 4, a supplementary table is more suitable.

Author Response

We sincerely thank Reviewer 1 for providing constructive feedback on the revised manuscript. We appreciate the comments provided to evolve our manuscript into it’s strongest version. Changes have been made as recommended.